# Development of a Swellable and Floating Gastroretentive Drug Delivery System (*sf*GRDDS) of Ciprofloxacin Hydrochloride

**DOI:** 10.3390/pharmaceutics15051428

**Published:** 2023-05-07

**Authors:** Yu-Kai Liang, Wen-Ting Cheng, Ling-Chun Chen, Ming-Thau Sheu, Hong-Liang Lin

**Affiliations:** 1School of Pharmacy, College of Pharmacy, Kaohsiung Medical University, Kaohsiung 80708, Taiwan; apccac1004@gmail.com; 2Department of Biotechnology and Pharmaceutical Technology, Yuanpei University of Medical Technology, Hsinchu 30015, Taiwan; wenting@mail.ypu.edu.tw (W.-T.C.); d8801004@mail.ypu.edu.tw (L.-C.C.); 3School of Pharmacy, College of Pharmacy, Taipei Medical University, Taipei 11031, Taiwan; mingsheu@tmu.edu.tw; 4Drug Development and Value Creation Research Center, Kaohsiung Medical University, Kaohsiung 80708, Taiwan

**Keywords:** gastroretentive drug delivery system, hydrophobically modified hydroxypropyl methylcellulose, Sangelose, ciprofloxacin, biphasic drug release, pharmacokinetics

## Abstract

Sangelose^®^ (SGL) is a novel hydroxypropyl methylcellulose (HPMC) derivative that has been hydrophobically modified. Due to its high viscosity, SGL has the potential as a gel-forming and release-rate-controlled material for application in swellable and floating gastroretentive drug delivery systems (*sf*GRDDS). The aim of this study was to develop ciprofloxacin (CIP)-loaded *sf*GRDDS tablets comprised of SGL and HPMC in order to extend CIP exposure in the body and achieve optimal antibiotic treatment regimes. Results illustrated that SGL-HPMC-based *sf*GRDDS could swell to a diameter above 11 mm and showed a short floating lag time (<4 s) and long total floating time (>24 h) to prevent gastric emptying. In dissolution studies, CIP-loaded SGL-HPMC *sf*GRDDS demonstrated a specific biphasic release effect. Among the formulations, the SGL/type-K HPMC 15,000 cps (HPMC 15K) (50:50) group exhibited typical biphasic release profiles, with F4-CIP and F10-CIP individually releasing 72.36% and 64.14% CIP within 2 h dissolution, and sustaining release to 12 h. In pharmacokinetic studies, the SGL-HPMC-based *sf*GRDDS demonstrated higher *C_max_* (1.56–1.73 fold) and shorter *T_max_* (0.67 fold) than HPMC-based *sf*GRDDS. Furthermore, SGL 90L in GRDDS indicated an excellent biphasic release effect and a maximum elevation of relative bioavailability (3.87 fold). This study successfully combined SGL and HPMC to manufacture *sf*GRDDS that retain CIP in the stomach for an optimal duration while improving its pharmacokinetic characteristics. It was concluded that the SGL-HPMC-based *sf*GRDDS is a promising biphasic antibiotic delivery system that can both rapidly achieve the therapeutic antibiotic concentration and maintain the plasma antibiotic concentration for an extended period to maximize antibiotic exposure in the body.

## 1. Introduction

The oral drug delivery system is recognized as the most convenient and costly expedient for daily usage [1]. However, many orally administrated active pharmaceutical ingredients (APIs) still have some unfavorable pharmacokinetic characterizations, such as poor oral bioavailability, short half-life in plasma and high metabolic clearance [2]. These defects usually lead to a decline in the potency of APIs and further influence the medication adherence of patients. To overcome these disadvantages, the gastroretentive drug delivery system (GRDDS) is one of the strategies successfully developed in the pharmaceutical industry [3,4].

GRDDS with perpetuated gastric residence time is of particular interest for drugs that are restrictedly active in the stomach (misoprostol, ranitidine, amoxicillin and metronidazole) [5,6]. GRDDS is also profitable for drugs that have a relatively narrow absorption window in the stomach or the upper part of the small intestine (e.g., levodopa, pregabalin, cilostazol and ciprofloxacin), drugs that easily tend to generate plasma fluctuations (e.g., clarithromycin and ciprofloxacin), drugs that have low solubility and poor absorption in the high pH-value condition (e.g., diazepam, atenolol and ciprofloxacin) and drugs that alter the normal flora of the colon (antibiotics, e.g., ciprofloxacin) [6,7]. In addition, GRDDS enables the retention of the formulations in the stomach through various mechanisms, including floating, swelling, bioadhesive, high-density and magnetic systems [3,8]. Multiple gastroretentive mechanisms in a formulation are able to maintain the gastric tract effectively. In previous studies, Lin et al. developed swelling and floating GRDDS (*sf*GRDDS) to prolong the dissolution and enhance the oral bioavailability of nilotinib [9]. Hydroxypropyl methylcellulose (HPMC), a water-soluble polymer derived from cellulose, is exerted as the swelling and floating agent due to its highly capable aqueous absorption and gas trapping in dual mechanism GRDDS [10,11]. Unfortunately, since HPMC has a relatively higher erosion rate by gastric fluids and poor mechanical strength by gastric peristalsis, it usually combines with other excipients to maintain the gastroretentive property, including polyethylene oxide (PEO) and hydroxyethyl cellulose (HEC) [9,12]. According to the Noyes–Whitney equation, the high molecular weight excipients may constitute the higher viscosity and thicker gel layer in an aqueous solution to decrease the erosion rate and sustain the drug release [13]. Hence, the higher molecular weight agent is the unmet need in GRDDS.

Sangelose^®^ (SGL) is a commercial HPMC derivative that has been hydrophobically modified by the introduction of a stearyl group. It is widely used in cosmetics as a thickening agent [14,15]. In recent studies, SGL acted as a gel-forming agent to enhance viscosity and mechanical strength [15,16,17]. It was also combined with cyclodextrin to construct thermoresponsive hydrogels to administer through injection or external use [14]. The structure of SGL is shown in Figure 1a. Due to hydrophobically modifying with the stearyl group, SGL has a higher viscosity and water retentivity than HPMC [15,17]. The presence of the stearyl oxy-hydroxypropoxy groups in SGL leads to the hydrophobic bonding between chemical molecules; the hydroxy group of water could not break these strong hydrophobic interactions easily, resulting in SGL possessing a sturdy gel structure [18]. Hence, SGL is a potential material for the adjuvant to *sf*GRDDS.

Ciprofloxacin hydrochloride (CIP), 1-cyclopropyl-6-fluoro-4-oxo-7-piperazin-1-ylquinoline-3-carboxylic acid hydrochloride, is the second generation of broad-spectrum fluoroquinolone antibiotics [19]. The chemical structure of CIP is shown in Figure 1b. Due to the ability to block bacterial DNA gyrase, CIP is either crucially utilized to prevent and treat *Bacillus anthracis*-induced anthrax or treat the lower respiratory tract, otitis, sinusitis and cystitis [20,21]. According to global statistics from recent studies, approximately 1.83 billion people and 1.1 billion livestock are facing such lethal risks of *Bacillus anthracis*-induced anthrax in the vulnerable domain [22]. Since *Bacillus anthracis*-induced anthrax is a potentially fatal infectious disease in humans and livestock, prevention and treatment are critical issues to curtail the mortality. From the perspective of infection prevention, sufficient exposure to prophylactic antibiotics is an important prerequisite for bacteria elimination [23]. Unfortunately, CIP has a short-term half-life (3–4 h) in clinical pharmacokinetic studies, resulting in hasty plasma exposure for the prophylactic antibiotic [24]. Furthermore, in terms of physiochemical characterizations, the water solubility of CIP is dependent on pH value, which in an acidic environment has a higher solubility than in a basic environment [25]. CIP is primarily absorbed in the proximal gastrointestinal tract [26]. Hence, the GRDDS is an advantageous strategy to enhance solubility and prolong the drug release to elevate the CIP body exposure via retainment in the stomach. Thus, CIP was selected as a model drug to examine the efficaciousness of various combinations of HPMC and SGL on the physical characteristics of resulting GRDDS.

The aim of the present study is to develop CIP-loaded *sf*GRDDS tablets comprised of SGL and HPMC. The first part of the study was to confirm the gastroretentive capability of SGL-HPMC *sf*GRDDS tablets through swelling and floating capacities. The second part aimed to evaluate the suitability of CIP loaded into SGL-HPMC *sf*GRDDS tablets. The physical characterizations of tablets, including weight variation, thickness and tensile strength, were well evaluated. Following in vitro dissolution, studies investigated the release characterizations, and further release kinetic models were assessed to comprehend the release mechanism of tablets. The in vivo pharmacokinetic studies of optimized formulations were then evaluated through orally administrated CIP-loaded GRDDS tablets in New Zealand rabbits to confirm the improvement of in vivo characterizations.

## 2. Materials and Methods

### 2.1. Materials

Ciprofloxacin hydrochloride (CIP) was obtained from Shangyu Jingxin Pharmaceutical Co., Ltd. (Shaoxing, China). Commercial CIP tablets, Ciproxin^®^, were purchased from Bayer Pharma AG (lot ITA5888, Leverkusen, Germany). Hydroxypropyl methylcellulose stearoxy ether (Sangelose^®^ SGL 60L containing 27.0–30.0% methoxy group, 7.0–11.0% hydroxypropoxy group and 0.3–0.6% stearyloxyhydroxypropoxy group and SGL 90L containing 21.5–24.0% methoxy group, 7.0–11.0% hydroxypropoxy group and 0.3–0.6% stearyloxyhydroxypropoxy group) were supplied from DAIDO Chemical Corporation (Osaka, Japan). K-type hydroxypropyl methylcellulose 4000 and 15,000 cps (HPMC 4K and 15K) and crospovidone were purchased from Shin-Etsu Chemical Co., Ltd. (Tokyo, Japan). Sodium bicarbonate was obtained from Natural Soda, LLC (Rifle, CO, USA). Microcrystalline cellulose (MCC 102) was purchased from Wei Ming Pharmaceutical Mfg. Co., Ltd. (Taipei, Taiwan). All the reagents used in experiments were either pharmaceutical or chromatographic grade.

### 2.2. Qualification of Gastroretentive Capability of SGL-HPMC sfGRDDS Tablets

#### 2.2.1. Preparation of SGL-HPMC *sf*GRDDS Tablets

The formulations of SGL-HPMC *sf*GRDDS tablets were prepared via the direct compression method, and the tablets were used to verify the gastroretentive effect of SGL [9]. In the manufacturing process, the main controlled-release matrix of tablets was HPMC 4K or 15K combined with SGL 60L or 90L. In addition, crospovidone and sodium bicarbonate were used as the disintegrant and gas-generating agent, respectively. We investigated the properties of SGL/HPMC-based *sf*GRDDS using colloidal formers at three different ratios (50:50, 50:100 and 50:150) and varying molecular weights of SGL and HPMC. The composition of formulations is shown in Table 1. The whole of the polymers and excipients (SGL, HPMC, crospovidone and sodium bicarbonate) were initially passed through a No. 40 mesh and further mixed in a plastic bag for 3 min. After all materials were mixed properly, tablets were manufactured by weight into an 8.5-mm-diameter die and compressed with 40 ± 2 N of force via a tableting press (JC-DR-D, JCMCO, Taiwan).

#### 2.2.2. Swelling Studies of SGL-HPMC *sf*GRDDS Tablets

Swelling studies were conducted via the USP dissolution apparatus II method (Vankel VK 7000, Varian, Cary, NC, USA). The paddle speed was maintained at 50 rpm, and the dissolution medium used was 900 mL of simulated gastric fluid consisting of 0.1 N HCl at pH 1.2, which was kept at 37.0 ± 0.5 °C. The swelling was represented as the water-absorbing capacity and could be determined by either the diameter increment or weight gain of swollen tablets. The tablets were withdrawn at predetermined intervals (0, 1, 2, 3, 4, 5, 6, 7 and 8 h) and surface droplets were immediately removed by wiping with a paper towel. The diameter of swelling tablets (D_st_) was measured, and the weight-gain swelling ratio (*W_r_*) was calculated through the following equations:$$W_{r}\;\left(\%\right)=\frac{W_{t}-W_{i}}{W_{i}}\times 100$$
where *W_i_* and *W_t_* demonstrate the initial weight of dry tablets and that of swollen tablets at time *t* (0, 1, 2, 3, 4, 5, 6, 7 and 8 h), respectively.

#### 2.2.3. Floating Capacities of SGL-HPMC *sf*GRDDS Tablets

Floating capacities were determined using the USP dissolution apparatus II method (Vankel VK 7000, Varian, Cary, NC, USA). The paddle speed was maintained at 50 rpm, and the dissolution medium used was 900 mL of simulated gastric fluid consisting of 0.1 N HCl at pH 1.2, which was kept at 37.0 ± 0.5 °C. Floating capacities were divided into two indices of floating lag time (FLT) and total floating time (TFT). Individually, floating lag time is defined as the duration that immersed tablets require to rise to the water surface. The total floating time is the period of tablets continuously floating on the water’s surface.

### 2.3. Suitability of CIP Loaded into SGL-HPMC sfGRDDS Tablets

#### 2.3.1. Preparation of CIP-Loaded SGL-HPMC *sf*GRDDS Tablets

The formulations of CIP-loaded SGL-HPMC *sf*GRDDS were tableted by the direct compression method [9]. After the evaluation of SGL-HPMC *sf*GRDDS formulations, 250 mg ciprofloxacin was added to the optimized formulations to conduct further evaluations. The composition of formulations is shown in Table 2. The whole of the API, polymers and excipients were initially passed through a No. 40 mesh and further mixed in the plastic bag for 3 min. After the all materials were mixed properly, tablets were manufactured by weight into a 9-mm-diameter die with the CIP content equivalent to 291 mg/tablet (equal to 250 mg ciprofloxacin) and compressed with 40 ± 2 N of force via a tableting press (JC-DR-D, JCMCO, Taiwan).

#### 2.3.2. Physical Characterizations

##### Weight Variation Test

Weight variation was measured by weighing 3 CIP-loaded tablets of each formulation via an electronic balance (ME204, Mettler Toledo, OH, USA) and calculating the average weight to compare with the theoretical weight.

##### Thickness

Thickness was determined by a digimatic micrometer (MDC-25SX, Mitutoyo, Kanagawa, Japan) for 3 CIP-loaded tablets in each formulation. The thickness enabled us to compare the variation of the tableting compress process.

##### Tensile Strength

The tensile strength of CIP-loaded tablets was estimated via a tensile strength testing machine (JKT04F, JING KOOU, Taiwan). The results were triplicated in each formulation. The tensile strength indicated the extent to which the tablet can cope with compression force.

#### 2.3.3. In Vitro Dissolution Studies

In vitro dissolution studies were performed using the USP dissolution apparatus II method (Vankel VK 7000, Varian, Cary, NC, USA). The paddle speed was maintained at 50 rpm, and the dissolution medium used was 900 mL of simulated gastric fluid consisting of 0.1 N HCl at pH 1.2, which was kept at 37.0 ± 0.5 °C. Five-milliliter aliquots were collected at predetermined intervals (0, 1, 2, 3, 4, 6, 8, 10, 12, 14 and 24 h), and the same volume of fresh dissolution medium was replaced simultaneously. The drug concentrations of withdrawn samples were analyzed via a UV/VIS spectrophotometer (V-730, JASCO Corporation, Tokyo, Japan) at a wavelength of 277 nm. The calibration curve of y = 0.1241x + 0.0015 was validated with an *R*^2^ value of 0.9999, and the concentration range was 0.15–15 μg/mL.

#### 2.3.4. Release Kinetic Models

Release kinetic models were analyzed by DDSolver add-in software (an add-in program for Microsoft Excel) to evaluate the release mechanism [27,28,29]. The dissolution profiles were fitted into various drug release models, including zero-order, first-order, Higuchi, Korsmeyer–Peppas and Hixson–Crowell models. The mathematical equations of kinetic models are described below [30]:

Zero-order model
$$Q=k_{0}t+Q_{0}\;$$

First-order model
$$Q=Q_{0}\times e^{k_{1}t}\;$$

Higuchi model
$$Q=k_{H}\times t^{\frac{1}{2}}\;$$

Korsmeyer–Peppas model
$$Q=k_{kp}\times t^{n}\;$$

Hixson–Crowell model
$$Q^{\frac{1}{3}}=k_{HC}+Q_{0}{}^{\frac{1}{3}}\;$$
where *Q* indicates the fraction of drug released at time *t* and the *Q*_0_ demonstrates the initial value of *Q*. The *k*_0_, *k*_1_, *k_H_*, *K_kp_* and *k_HC_* represent the release rate constant of the various kinetic models, respectively. The *n* value of the Korsmeyer–Peppas model is the diffusion exponent and could indicate the releasing mechanism of formulations.

#### 2.3.5. In Vivo Pharmacokinetics

##### Animals

In vivo pharmacokinetic studies were performed on male New Zealand rabbits (body weight 3–4 kg) due to the pH value of the gastrointestinal environment being similar to humans [31]. All pharmacokinetic experiments in this study were approved by the Institutional Animal Care and Use Committee of Kaohsiung Medical University Hospital, Kaohsiung, Taiwan (approval no. IACUC-111127). The rabbits were subjected to a 12-h fasting period with ad libitum access to drinking water prior to drug administration. A single dose of 291 mg of the CIP formulation (equivalent to 250 mg of ciprofloxacin) was administered orally to the rabbits.

##### Experimental Procedure

The rabbits were divided into 4 groups: (a) commercial tablet (Ciproxin^®^); (b) F4-CIP; (c) F10-CIP; (d) F16-CIP. Each group contained 3 rabbits who were orally administrated the formulations containing 291 mg CIP (equaled 250 mg ciprofloxacin). Blood samples were withdrawn at 0, 0.5, 1, 1.5, 2, 4, 6, 8 and 12 h through the marginal ear vein of rabbits after oral tablet administration. Blood samples were collected into heparin-coated tubes and centrifuged at 4000 rpm for 10 min. After centrifuging, the supernatants of each sample were collected separately, and further plasma extraction processes were conducted.

##### Blood Sample Preparation

The blood samples adopted the protein precipitation method to extract the CIP from plasma. The protein precipitation was conducted by acetonitrile (ACN) [32]. In brief, a 100 µL plasma sample was added into 400 µL ACN to vortex for 3 min. After being vortexed, the samples were sonicated for 10 min and further centrifuged at 12,500 rpm for 10 min. After the above procedures, the supernatants were withdrawn and dried with a vacuum dryer under 75 °C for 60 min. The dried samples were reconstituted with 2% acetic acid to 100 µL and subjected to analysis via high-performance liquid chromatography (HPLC).

##### Chromatography

The plasma samples were analyzed and quantified via HPLC. The JASCO HPLC system (JASCO Corporation, Tokyo, Japan) was utilized; it comprised a PU-4180 pump, AS-4150 autosampler, UV-4075 UV/Vis detector and LC-NET II/ADC interface box. The stationary phase employed a LiChrospher^®^ 100 RP-18 column (250 × 4.6 mm, 5 µm; Merck KGaA, Darmstadt, Germany) to separate samples. The mobile phase consisted of 0.02 M disodium hydrogen phosphate buffer (adjusted to pH 2.7 with phosphoric acid) and acetonitrile (ACN) at the ratio of 80:20 (*v*/*v*) through isocratic elution, and the flow rate was controlled at 1.5 mL per min. The injection volume of samples was set at 40 µL, and the detection wavelength was situated at 277 nm. The calibration curve of y = 117633x − 9894.8 was validated with an *R*^2^ value of 0.9998, and the concentration range was 0.05–5 μg/mL.

### 2.4. Statistical Analysis

All experimental data are presented as the mean ± standard deviation (SD). The determination of statistical significance between means was performed by one-way analysis of variance (ANOVA). The value of *p* < 0.05 demonstrated statistical significance.

## 3. Results and Discussion

### 3.1. Qualification of Gastroretentive Capability of SGL-HPMC sfGRDDS Tablets

The aim of the present study is to develop CIP-loaded *sf*GRDDS tablets comprised of SGL and HPMC to elongate the period of gastric retention by curtailing the floating lag time, prolonging the floating duration and further sustaining the dissolution release pattern, proving the in vivo characterizations through a rabbit pharmacokinetic model. To attain the goal of elongating the period of gastric retention, one method is floating on the gastric fluid in the stomach immediately and for a substantial period, since it could prevent the risk of gastric emptying; the method other is swelling of the tablets to over about 11 mm diameter to avoid gastric emptying through the pyloric sphincter [33].

#### 3.1.1. Swelling Studies of SGL-HPMC *sf*GRDDS Tablets

All the SGL-HPMC *sf*GRDDS tablets experienced more than a 3-fold increase in weight-gain swelling ratio (*W_r_*) and about 1.3-fold increase of diameter after 8 h of sinking in the dissolution medium. The capacity of water absorption in GRDDS is an imperative mechanism that could produce swelling to avoid gastric emptying through the pylorus [6,34]. Plots of *W_r_* measurements are shown in Figure 2. The maximum 8-h *W_r_* ranged from 314.5 to 421.1%. This demonstrated whether SGL 60L or 90L combined with HPMC had the capacity of water absorption. Similar results were related to the diameter of swelling tablets (D_st_). Plots of D_st_ measurements are presented in Figure 3. The maximum 8-h D_st_ to un-swelling tablets ranged from 137.6 to 162.6%. This also indicated the improvement of the water absorption capacity of SGL-HPMC *sf*GRDDS tablets. The swelling mechanism of SGL-HPMC tablets is dominated by the interaction of intra- and intermolecular stearyl groups and hydroxy groups on SGL and HPMC [16,35]. Furthermore, it is noteworthy that within each group, the swelling capacity of formulations with a high ratio of SGL (F1, F4, F7, F10) was greater than those with a low ratio of SGL (F3, F6, F9, F12). Based on this observation, it can be inferred that because SGL dominates the water uptake ability, as the proportion of SGL increases, it is easier for water to penetrate the tablet, resulting in a greater degree of swelling. The results are potentially supported by a previous study that indicated that SGL had a lower viscosity in low concentration versus HPMC [36]. The potential mechanism for the fast-swelling behavior of SGL includes the long alkyl chain, which interrupts gel formation during the initial phase and facilitates rapid water penetration into the tablets. Conversely, as the proportion of SGL decreases, the impact of HPMC on the swelling property within the formulation is intensified, leading to the formation of a more stable gel structure in the tablet, which reduces water permeability and results in a weaker swelling ability.

Gao et al. investigated the swelling characterizations of HPMC-based tablets with various proportions and different types of HPMC [37]. In this study, HPMC-based tablets were composed of API (adinazolam mesylate), HPMC, lactose and magnesium stearate. The results demonstrated that the diameter of tablets was maintained below the initial length (8 mm) after 8 h immersion in an aqueous condition. Nevertheless, in our presented study, the addition of SGL and sodium bicarbonate in tablets could result in swelling and floating, allowing the tablets to stay in the stomach and sustaining the drug release.

#### 3.1.2. Floating Capacities of SGL-HPMC *sf*GRDDS Tablets

The FLT indicated the ability to refloat tablets, while the longer TFT represented the sustainability of the floating capacity of tablets. The data of FLT and TFT are presented in Table 3. The FLT and TFT of all the SGL-HPMC *sf*GRDDS tablets were less than 4 s and steadily more than 24 h, respectively. The lower FLT and longer TFT could lower the risk of gastric emptying after swallowing the tablets into the stomach. In terms of the results, the sodium bicarbonate would generate carbon dioxide bubbles when encountered in the hydrochloric acid solution. Further, the bubbles were immediately entrapped in hydrated SGL and HPMC, resulting in a short period of FLT and a long duration of TFT. In summary, SGL combined with HPMC in GRDDS exhibited stable performance in swelling and floating capacities. The formulations were loaded with CIP to evaluate subsequent dissolution properties.

### 3.2. Suitability of CIP Loaded into SGL-HPMC sfGRDDS Tablets

After verifying the gastroretentive effect of SGL combined with HPMC in *sf*GRDDS, CIP was loaded into SGL-HPMC *sf*GRDDS tablets to evaluate the in vitro dissolution and in vivo pharmacokinetic properties. Through dissolution and pharmacokinetic studies, the further release efficacy could be established and the role of SGL in *sf*GRDDS could be clarified.

#### 3.2.1. Physical Characterization and Floating Capacities

The physical characterization and floating capacities are illustrated in Table 4. In the weight variation test, all formulations exhibited slight fluctuations (<4%) compared to theoretical tablet weight. This indicated that SGL was an adequate excipient in tablet manufacturing. The thickness of the tablets indicated an increment over the tablet weight due to the enlargement of feeding powder in the tablet mold. Moreover, the tensile strength demonstrated >114 N (114.67–142.97 N) of the formulations. A similar strength of floating tablets was shown in previous studies [38]. This demonstrated a mechanical strength in all formulations within an appropriate range. Interestingly, the formulation consisting of SGL or HPMC manifested lower mechanical strength compared to single MCC-constructed formulations. One potential reason is the low-density excipient, crospovidone, introduced into tablets, resulting in the fluffy structure of tablets [39]. Although crospovidone potentially decreased the mechanical strength of tablets, it still played an important role in *sf*GRDDS, facilitating water uptake into tablets to accelerate gel formation [40,41]. Moreover, despite MCC having higher tensile strength, the formulations lacked the viscous polymeric gelling agents, resulting in quick disintegration and burst-releasing characteristics [42]. In floating capacity, the hydrophilic polymer-based formulations exhibited the same gastroretentive effect after loading 291 mg CIP over non-CIP-loaded formulations. In addition, the formulations of the MCC group prove that CIP does not influence the floating capacities.

#### 3.2.2. In Vitro Dissolution Studies

The dissolution profiles of the CIP-loaded *sf*GRDDS tablets are shown in Figure 4. The results notably exhibited the particular biphasic release mode of SGL in HPMC-based *sf*GRDDS. In a recent study, SGL-based gel was reported to have lower viscosity than HPMC-based gel at low polymer concentration; conversely, SGL-based gel had higher viscosity versus HPMC-based gel at high polymer concentration [36]. The gel viscosity directly influences the gel strength, which controls the release rate of the drug. The higher viscosity gel lessens the drug release from the matrix to decelerate the dissolution [43,44]. These results support that SGL in *sf*GRDDS has a biphasic release effect. Furthermore, the biphasic release effect is decreased with the increment of the HPMC amount. This study demonstrated that HPMC forms a high-viscosity gel layer when it initially encounters water, resulting in a decrease in the initial release characteristics.

The dissolution results of the formulations composed of SGL/HPMC 4K (50:50) are shown in Figure 4a. The F1-CIP and F7-CIP demonstrated burst release (>75%) within 2 h compared to F13-CIP. The formulations composed of SGL/HPMC 15K (50:50) are illustrated in Figure 4b. They indicate that the burst effect was slowed in F4-CIP and F10-CIP due to higher molecular HPMC producing a sturdy gel layer to prevent the drug release from the matrix. With the increment of the HPMC amount, the dissolution profiles were similar, as the dissolution rate was dominated by HPMC. Nevertheless, SGL-based formulations still exhibited biphasic release characteristics, as seen in Figure 4c–f. The formulations containing MCC showed that CIP was rapidly released from the tablets within 2 h, while the commercial tablets exhibited a similar rapid release profile within 1 h. This demonstrates that CIP can be released quickly without any GRDDS matrix to retard drug release. The MCC-loaded formulations and commercial tablets are illustrated in Figure 4g,h, respectively.

The dissolution parameters of the CIP-loaded *sf*GRDDS tablets are shown in Appendix A. Among the various parameters, mean dissolution time (MDT) was used for drug release rate and retarding efficiency of the polymeric matrix, and dissolution efficiency (DE) was the ratio of the area under the dissolution curves over the area of 100% release and represented the drug release efficiency [45,46]. From the results of Q_2h_, Q_12h_ and Q_24h_ in each group, the biphasic release effect was also observed on SGL-HPMC *sf*GRDDS formulations. MDT values were increased with the accruement of the HPMC amount. In contrast, the DE value was decreased with the increment of the HPMC amount due to a lower ratio of hydrophilic polymers in the tablet matrix accelerating the dissolution rate. To sum up, the results of MDT and DE supported that the molecular weight of HPMC-related polymers influenced the dissolution, with the higher molecular weight showing a decreased dissolution rate [37,47].

Gaikwad, V. D. et al. developed CIP-loaded *sf*GRDDS, and the formulations were mainly composed of different molecular weight HPMC and disintegrants [7]. Further, the formulations of CIP-loaded *sf*GRDDS demonstrated that had a higher FLT and lower TFT led to dissolution of the CIP at 12 h. In our presented study, the addition of a high viscosity polymer (SGL 60L or 90L) to *sf*GRDDS could facilitate rapid release at the initial dissolution phase and also improve the sustained release effect of dissolution up to 24 h.

In summary, SGL demonstrated the ability of biphasic release of CIP in HPMC-based GRDDS. In addition to the high viscosity of SGL materials providing high gel strength in the GRDDS structure, the particular biphasic releasing effect was noticed; the immediate release supplied quick drug concentration to deliver therapeutic effects within a rapid period, and further extended release contributed to maintaining drug concentration with slight fluctuation and prolonged the therapeutic effects. The SGL-HPMC 15K (50:50) group exhibited the typical biphasic release character; pharmacokinetic studies were evaluated to clarify the relationship of pharmacokinetic parameters between different hydrophilic polymer (SGL and HPMC) compositions.

#### 3.2.3. Release Kinetic Models

The release kinetics of dissolution were evaluated by fitting five mathematical models, including zero-order, first-order, Higuchi, Korsmeyer–Peppas and Hixson-Crowell models. The best-fitting release model was chosen using the coefficient of determination (*R*^2^), where the value of *R*^2^ close to 1 represented good fit, while an *R*^2^ value below 0 demonstrated poor fit [48,49]. According to the *R*^2^ formula, a negative value indicated the that regression model performed inadequately and was an inappropriate fit [50]. The release kinetic mathematical models are shown in Table 5.

From the results, F1-CIP, F4-CIP and F7-CIP, consisting of lower molecular weight polymers (SGL 60L or HPMC 4K), were demonstrated as the best fit (*R*^2^ = 0.8671, 0.9779 and 0.8634, respectively) of the first-order model. This indicated the release rate of F1-CIP, F4-CIP and F7-CIP was dependent on the concentration of CIP. F2-CIP, F3-CIP, F5-CIP, F6-CIP, F8-CIP, F9-CIP, F10-CIP, F11-CIP and F12-CIP were fitted to the Korsmeyer–Peppas model (*R*^2^ = 0.9933, 0.9963, 0.9967, 0.9977, 0.9980, 0.9977, 0.9126, 0.9930 and 0.9983, respectively). In the Korsmeyer–Peppas model, the value of *n* represented the mechanism of release. An *n* value below 0.45 indicated the diffusion-type drug release mechanism, and an *n* value above 0.89 indicated the swelling-type drug release mechanism. However, an *n* value between 0.45 to 0.89 would be considered as the dual release mechanism (anomalous diffusion), containing diffusion and swelling [30,51]. Among these formulations, the *n* value of F2-CIP, F5-CIP and F10-CIP, respectively, was 0.418, 0.443 and 0.197, demonstrating that diffusion was the main release mechanism. In F2-CIP and F5-CIP, the lower molecular weight of SGL formed a loose gel layer that would not entrap the CIP, effectively resulting in drug release controlled by a diffusion mechanism. F10-CIP facilitated the diffusion-releasing mechanism due to the lower content of gelling polymers, which could not construct a sufficient gelling layer to snare the CIP. Furthermore, the *n* value of F3-CIP, F6-CIP, F8-CIP, F9-CIP, F11-CIP and F12-CIP (0.556, 0.604, 0.464, 0.578, 0.546 and 0.629, respectively) was between 0.45 and 0.89, which indicated that both diffusion and swelling were the mechanisms controlling drug release. Compared with F2-CIP and F5-CIP, F8-CIP and F11-CIP had the same amount of gelling polymers, but the higher molecular weight of SGL predominated the swelling-releasing mechanism via generating the robust gelling layer to entrap the CIP release. Meanwhile, F3-CIP, F6-CIP, F9-CIP and F12-CIP comprised a high content of hydrophilic polymers, which originated from the thick hydrate gel layers. It could ensnare the CIP release from the polymeric gel layer and further contribute to the release mechanisms of diffusion and swelling. In addition, the *n* value in each SGL-HPMC formulation group exhibited a correlation with the concentration of HPMC. When the concentration of HPMC increased in each formulation group, the *n* value was also increased simultaneously. This indicated that a higher hydrophilic polymer content in *sf*GRDDS systems could form stauncher gelling layers, influencing the release kinetics through the swelling mechanism.

Furthermore, compared with the pure HPMC group under the Korsmeyer–Peppas model, the rate constants *K_KP_* of the SGL 60L/HPMC 4K, SGL 90L/HPMC 4K and HPMC 4K groups decreased in the order of SGL60>SGL90L>non-SGL, which confirmed the results of the previous dissolution studies. SGL demonstrated efficient early-stage rapid drug release, resulting in a faster release rate for the SGL group compared to the HPMC group. Additionally, the SGL 60L group was faster than the SGL 90L group due to the lower molecular weight of SGL 60L, which made it easier for the drug to be rapidly released in the early stages and prevented the formation of a sturdy gel structure, thereby increasing its release capacity compared to SGL 90L. Similar results were observed in the SGL 60L/HPMC 15K, SGL 90L/HPMC 15K and HPMC 15K groups.

#### 3.2.4. In Vivo Pharmacokinetics

Pharmacokinetic studies were conducted to comprehend the in vivo characteristics of SGL in GRDDS. The plasma concentration–time curves after oral administration of 250 mg ciprofloxacin commercial tablets and *sf*GRDDS formulations are shown in Figure 5. The commercial tablets demonstrated the lowest concentration profile due to the rapid gastric emptying in the fasted rabbits without any precaution. The F16-CIP combined with HPMC 15K had a lag release period of 1 h. The similar lag pharmacokinetic profiles of HPMC-based GRDDS were also demonstrated in a previous study [26]. HPMC forming a high viscosity gel layer on the tablet surface prevented water from penetrating the tablets and resulted in a lag of the initial release effect. In contrast, the SGL-HPMC-based *sf*GRDDS did not exhibit lag in the initial release due to the initial low viscosity of the SGL gel layer; it demonstrated at rapid release in the inceptive dissolution phase and also elevated the plasma drug concentration in the early digestion stage. Pharmacokinetic parameters of oral administration of 250 mg ciprofloxacin commercial tablets and *sf*GRDDS formulations are illustrated in Table 6. The *C_max_* values of F4-CIP and F10-CIP (2.49 ± 0.20 and 2.77 ± 0.38 μg/mL) were significantly higher than those of the commercial tablets and F19-CIP (0.86 ± 0.24 and 1.60 ± 0.36 μg/mL). The *T_max_* of commercial tablets F4-CIP and F10-CIP (1.33 ± 0.29 h) were shorter than F16-CIP (2.0 ± 0.0 h). There was no statistically significant difference in the MRT values among the groups, but F4-CIP exhibited the shortest MRT due to its weak gel strength, which resulted in ineffective entrapment of CIP, causing high initial burst release and an inability to sustain the release of CIP. The AUC_0-t_ of F10-CIP (12.78 ± 3.25 μg/mL/h) was significantly higher than that of the commercial tablets (3.30 ± 0.63 μg/mL/h). Furthermore, the AUC0-∞ of F10-CIP and F16-CIP (14.33 ± 4.28 and 13.21 ± 4.33 μg/mL/h, respectively) was significantly higher than that of the commercial tablets (4.72 ± 0.85 μg/mL/h).

Compared to commercial tablets, F4-CIP, F10-CIP and F16-CIP demonstrated elevated relative bioavailability by about 2.83–4.79 fold due to rapid gastric emptying of commercial tablets, resulting in poor drug absorption in the lower part of the GI tract. Among F4-CIP, F10-CIP and F16-CIP, F10-CIP exhibited the highest increase of relative bioavailability (387.33 ± 98.43%), which was attributed to the crucial biphasic drug release characteristics of SGL. Although F4-CIP was also composed of SGL, F4-CIP was limited to an increase in relative bioavailability of 238.35 ± 45.94%, since SGL 60L had a relatively lower molecular weight than SGL 90L, leading to rapid release rather than effectually entrapping the drug release due to the gel layer. A similar value of relative bioavailability was shown for the F16-CIP group, which demonstrated 247.34 ± 45.46% relative bioavailability compared to the commercial tablets. From the perspective of pharmacokinetic curves, F4-CIP manifested rapid absorption and elimination; nevertheless, F16-CIP exhibited a lag in absorption at the initial phase and maintained the plasma concentration for 12 h. Though F4-CIP and F16-CIP demonstrated comparable relative bioavailability, they possessed different pharmacokinetic characteristics. In our pharmacokinetic studies, *sf*GRDDS formulations raised AUC values and improved relative bioavailability compared to non-gastroretentive commercial tablets, indicating that *sf*GRDDS is a promising strategy to ameliorate pharmacokinetic characteristics. Furthermore, from the pharmacokinetic parameters, the SGL-HPMC-based *sf*GRDDS demonstrated significantly higher *C_max_* (1.56–1.73 fold) and shorter *T_max_* (0.67 fold) than HPMC-based *sf*GRDDS. It contributed to the particular biphasic release effect of SGL. Although SGL had the capability of rapid release in the initial dissolution phase, SGL 90L exhibited an excellent sustained release effect compared to SGL 60L, suggesting that high molecular weight SGL could form the higher viscosity gel layer to entrap the CIP release from the tablets.

## 4. Conclusions

This study successfully combined hydrophobically modified HPMC (SGL) and HPMC, loaded to *sf*GRDDS, in order to retain CIP in the stomach for an optimal duration while improving its pharmacokinetic characteristics. The SGL-HPMC-loaded *sf*GRDDS exhibited excellent swelling and floating capacities while simultaneously possessing the unique biphasic drug release potential. In terms of pharmacokinetics, we verified that the SGL-HPMC *sf*GRDDS could shorten the initial lag onset of drug concentration and improve the relative bioavailability. Due to the specific biphasic drug release feature of SGL, the formulations effectively prolonged the exposure of CIP in vivo. Overall, the SGL-HPMC-based *sf*GRDDS represents a promising biphasic-released antibiotic delivery system that could rapidly achieve therapeutic antibiotic concentrations while maintaining plasma antibiotic concentration for an extended period, thereby maximizing exposure in the body and optimizing therapeutic efficacy.

## Figures and Tables

**Figure 1 pharmaceutics-15-01428-f001:**
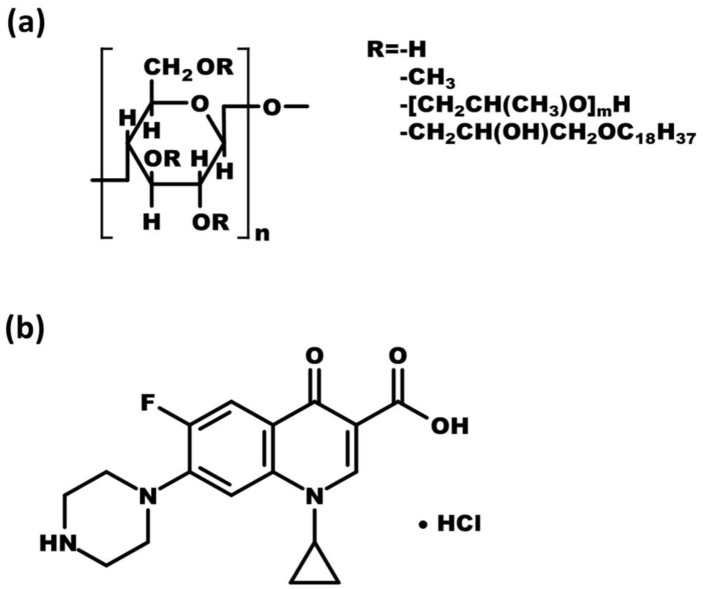
Chemical structure of (**a**) SGL and (**b**) CIP.

**Figure 2 pharmaceutics-15-01428-f002:**
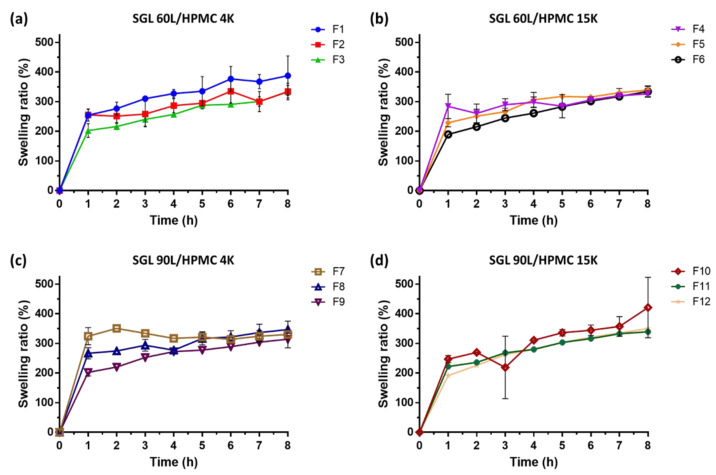
Profiles of *W_r_* measurements of (**a**) SGL 60L/HPMC 4K, (**b**) SGL 60L/HPMC 15K, (**c**) SGL 90L/HPMC 4K and (**d**) SGL 90L/HPMC 15K SGL-HPMC tablets. Each data point represents mean ± standard error of the mean (*n* = 3 for each group).

**Figure 3 pharmaceutics-15-01428-f003:**
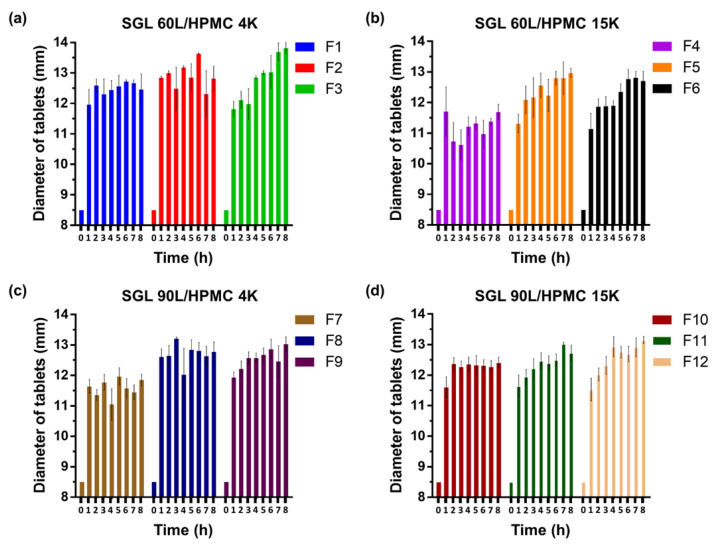
Plots of D_st_ measurements of (**a**) SGL 60L/HPMC 4K, (**b**) SGL 60L/HPMC 15K, (**c**) SGL 90L/HPMC 4K and (**d**) SGL 90L/HPMC 15K SGL-HPMC tablets. Each data point represents mean ± standard error of the mean (*n* = 3 for each group).

**Figure 4 pharmaceutics-15-01428-f004:**
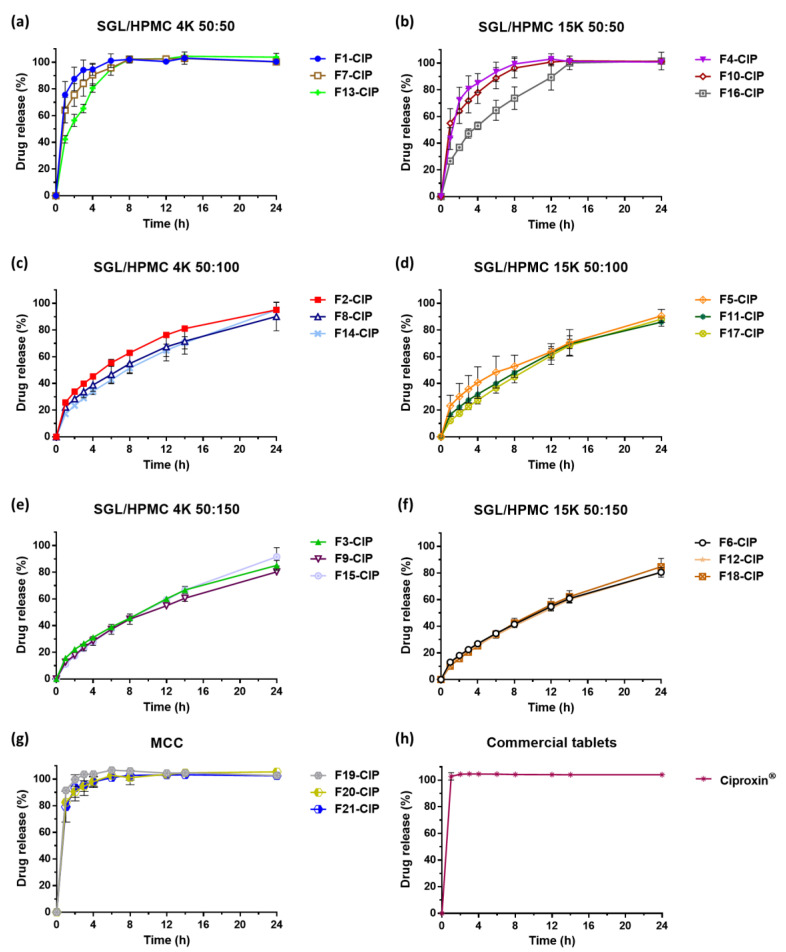
Dissolution profiles of CIP-*sf*GRDDS tablets composed of (**a**) SGL/HPMC 4K (50:50), (**b**) SGL/HPMC 15K (50:50), (**c**) SGL/HPMC 4K (50:100), (**d**) SGL/HPMC 15K (50:100), (**e**) SGL/HPMC 4K (50:150), (**f**) SGL/HPMC 15K (50:150), (**g**) MCC and (**h**) commercial tablets in 0.1 N HCl. Each data point represents mean ± standard error of the mean (*n* = 3 for each group).

**Figure 5 pharmaceutics-15-01428-f005:**
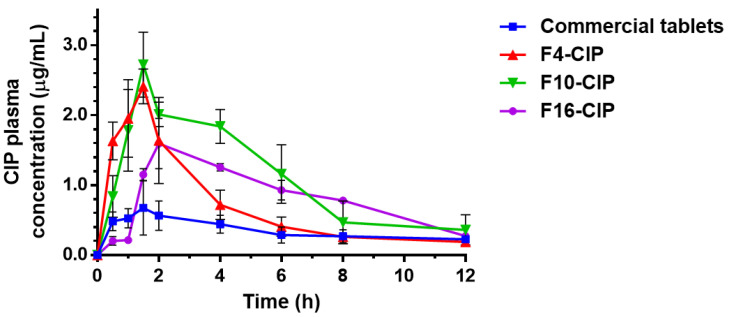
Pharmacokinetic profiles of CIP plasma concentration over time after oral administration of 250 mg ciprofloxacin commercial tablets and *sf*GRDDS formulations (F4-CIP, F10-CIP and F16-CIP). Each data point represents mean ± standard error of the mean (*n* = 3 for each group).

**Table 1 pharmaceutics-15-01428-t001:** Composition of each SGL-HPMC *sf*GRDDS tablet.

Ingredients (mg/per Tablet)	SGL 60L/HPMC 4K			SGL 60L/HPMC 15K			SGL 90L/HPMC 4K			SGL 90L/HPMC 15K		
	F1	F2	F3	F4	F5	F6	F7	F8	F9	F10	F11	F12
SGL 60L	50	50	50	50	50	50	-	-	-	-	-	-
SGL 90L	-	-	-	-	-	-	50	50	50	50	50	50
HPMC 4K	50	100	150	-	-	-	50	100	150	-	-	-
HPMC 15K	-	-	-	50	100	150	-	-	-	50	100	150
Crospovidone	150	150	150	150	150	150	150	150	150	150	150	150
NaHCO_3_	50	50	50	50	50	50	50	50	50	50	50	50
Total weight	300	350	400	300	350	400	300	350	400	300	350	400

Abbreviations: SGL: hydroxypropyl methylcellulose stearoxy ether; HPMC: hydroxypropyl methylcellulose.

**Table 2 pharmaceutics-15-01428-t002:** Composition of each CIP-loaded *sf*GRDDS tablet.

Ingredients (mg/per Tablet)	SGL 60L/ HPMC 4K			SGL 60L/ HPMC 15K			SGL 90L/ HPMC 4K			SGL 90L/ HPMC 15K			HPMC 4K			HPMC 15K			MCC		
	F1-CIP	F2-CIP	F3-CIP	F4-CIP	F5-CIP	F6-CIP	F7-CIP	F8-CIP	F9-CIP	F10-CIP	F11-CIP	F12-CIP	F13-CIP	F14-CIP	F15-CIP	F16-CIP	F17-CIP	F18-CIP	F19-CIP	F20-CIP	F21-CIP
CIP ^1^	291	291	291	291	291	291	291	291	291	291	291	291	291	291	291	291	291	291	291	291	291
SGL 60L	50	50	50	50	50	50	-	-	-	-	-	-	-	-	-	-	-	-	-	-	-
SGL 90L	-	-	-	-	-	-	50	50	50	50	50	50	-	-	-	-	-	-	-	-	-
HPMC 4K	50	100	150	-	-	-	50	100	150	-	-	-	100	150	200	-	-	-	-	-	-
HPMC 15K	-	-	-	50	100	150	-	-	-	50	100	150	-	-	-	100	150	200	-	-	-
Crospovidone	150	150	150	150	150	150	150	150	150	150	150	150	150	150	150	150	150	150	-	-	-
NaHCO_3_	50	50	50	50	50	50	50	50	50	50	50	50	50	50	50	50	50	50	-	-	-
MCC	-	-	-	-	-	-	-	-	-	-	-	-	-	-	-	-	-	-	300	350	400
Total weight	591	641	691	591	641	691	591	641	691	591	641	691	591	641	691	591	641	691	591	641	691

^1^ 291 mg CIP is equal to 250 mg ciprofloxacin.

**Table 3 pharmaceutics-15-01428-t003:** Floating capacities of SGL-HPMC *sf*GRDDS tablets.

Floating Capacities	F1	F2	F3	F4	F5	F6	F7	F8	F9	F10	F11	F12
FLT ^1^ (s)	4 ± 1	3 ± 1	2 ± 1	2 ± 1	2 ± 1	2 ± 1	2 ± 1	2 ± 1	2 ± 1	2 ± 1	2 ± 1	2 ± 1
TFT ^2^ (h)	>24	>24	>24	>24	>24	>24	>24	>24	>24	>24	>24	>24

^1^ FLT = floating lag time; ^2^ TFT = total floating time. Data represent mean ± standard error of the mean (*n* = 3 for each group).

**Table 4 pharmaceutics-15-01428-t004:** Physical characterizations and floating capacities of CIP-loaded *sf*GRDDS tablets.

Formulations	Average Weight (mg)	Thickness (mm)	Tensile Strength (N)	Floating Capacities ^1^		
				6 h	12 h	24 h
F1-CIP	601.70 ± 0.79	9.69 ± 0.01	115.00 ± 0.61	ο	ο	ο
F2-CIP	646.20 ± 7.22	10.06 ± 0.09	120.83 ± 1.94	ο	ο	ο
F3-CIP	696.80 ± 6.22	10.69 ± 0.02	123.60 ± 0.35	ο	ο	ο
F4-CIP	599.13 ± 1.61	9.24 ± 0.06	122.63 ± 1.29	ο	ο	ο
F5-CIP	651.90 ± 3.12	10.04 ± 0.10	121.03 ± 2.38	ο	ο	ο
F6-CIP	705.83 ± 0.68	10.91 ± 0.09	119.47 ± 1.62	ο	ο	ο
F7-CIP	602.27 ± 4.77	9.33 ± 0.17	121.93 ± 2.87	ο	ο	ο
F8-CIP	644.77 ± 6.91	9.88 ± 0.12	124.77 ± 0.68	ο	ο	ο
F9-CIP	703.67 ± 0.31	10.64 ± 0.08	124.40 ± 1.28	ο	ο	ο
F10-CIP	595.37 ± 5.72	9.73 ± 0.18	114.67 ± 3.16	ο	ο	ο
F11-CIP	640.83 ± 3.21	9.76 ± 0.06	126.03 ± 0.76	ο	ο	ο
F12-CIP	694.87 ± 10.12	10.60 ± 0.07	123.47 ± 1.00	ο	ο	ο
F13-CIP	598.27 ± 1.32	9.36 ± 0.12	122.27 ± 2.06	ο	ο	ο
F14-CIP	643.57 ± 2.84	9.79 ± 0.05	125.53 ± 0.83	ο	ο	ο
F15-CIP	695.93 ± 2.05	10.64 ± 0.01	120.87 ± 5.00	ο	ο	ο
F16-CIP	596.30 ± 1.06	9.21 ± 0.11	124.07 ± 0.78	ο	ο	ο
F17-CIP	643.10 ± 0.78	9.64 ± 0.05	127.10 ± 0.36	ο	ο	ο
F18-CIP	695.13 ± 5.50	10.56 ± 0.04	124.80 ± 1.93	ο	ο	ο
F19-CIP	588.40 ± 0.96	7.79 ± 0.01	142.97 ± 0.57	×	×	×
F20-CIP	638.40 ± 0.20	8.45 ± 0.25	142.97 ± 3.00	×	×	×
F21-CIP	689.13 ± 0.40	8.98 ± 0.01	142.63 ± 0.91	×	×	×

^1^ ο = floating status; × = sunken status. Data represent mean ± standard error of the mean (*n* = 3 for each group).

**Table 5 pharmaceutics-15-01428-t005:** Parameters of release kinetic models of CIP-loaded *sf*GRDDS tablets.

Groups	Formulations	Zero-order		First-order		Higuchi		Korsmeyer–Peppas			Hixson–Crowell	
		*K* _0_	*R* ^2^	*K* _1_	*R* ^2^	*K_H_*	*R* ^2^	*K_KP_*	*n*	*R* ^2^	*K_HC_*	*R* ^2^
SGL 60L/ HPMC 4K	F1-CIP	7.069	−46.0369	1.264	0.8671	31.269	−14.9395	83.114	0.080	0.7500	0.070	−14.7455
	F2-CIP	5.318	0.0141	0.141	0.9183	21.355	0.9636	26.039	0.418	0.9933	0.039	0.8463
	F3-CIP	4.361	0.6620	0.083	0.9620	16.955	0.9883	14.778	0.556	0.9963	0.023	0.9291
SGL 60L/ HPMC 15K	F4-CIP	6.882	−6.0014	0.567	0.9779	29.662	−0.7998	61.733	0.190	0.7832	0.068	−0.4621
	F5-CIP	4.773	0.2028	0.106	0.8514	18.991	0.9839	21.808	0.443	0.9967	0.029	0.7655
	F6-CIP	4.020	0.7694	0.070	0.9768	15.475	0.9741	11.969	0.604	0.9977	0.020	0.9525
SGL 90L/ HPMC 4K	F7-CIP	6.976	−15.4187	0.785	0.8634	30.431	−3.7967	71.680	0.134	0.8282	0.069	−3.4228
	F8-CIP	4.811	0.3180	0.107	0.9113	19.080	0.9934	20.805	0.464	0.9980	0.030	0.8446
	F9-CIP	4.060	0.7156	0.073	0.9723	15.723	0.9834	12.990	0.578	0.9977	0.021	0.9373
SGL 90L/ HPMC 15K	F10-CIP	6.767	−6.3842	0.497	0.8403	28.943	−0.7270	59.166	0.197	0.9126	0.067	−0.4322
	F11-CIP	4.476	0.6313	0.088	0.9657	17.448	0.9874	15.580	0.546	0.9930	0.025	0.9320
	F12-CIP	3.964	0.8120	0.068	0.9827	15.183	0.9648	11.041	0.629	0.9983	0.020	0.9649
HPMC 4K	F13-CIP	6.927	−2.8559	0.429	0.9570	29.363	0.1929	53.245	0.250	0.8384	0.067	0.4487
	F14-CIP	4.779	0.6861	0.097	0.9653	18.544	0.9892	15.851	0.564	0.9994	0.027	0.9465
	F15-CIP	4.447	0.8544	0.081	0.9873	16.937	0.9523	11.419	0.659	0.9989	0.023	0.9869
HPMC 15K	F16-CIP	6.064	−0.1063	0.198	0.9652	24.529	0.9121	30.884	0.405	0.9537	0.054	0.9406
	F17-CIP	4.404	0.8226	0.081	0.9915	16.843	0.9551	11.910	0.640	0.9933	0.023	0.9858
	F18-CIP	4.133	0.8604	0.072	0.9938	15.722	0.9477	10.427	0.665	0.9974	0.021	0.9876

**Table 6 pharmaceutics-15-01428-t006:** Pharmacokinetic parameters of oral administration of 250 mg ciprofloxacin commercial tablets and *sf*GRDDS formulations (F4-CIP, F10-CIP and F16-CIP).

Pharmacokinetic Parameters	Commercial Tablets	SGL 60L/ HPMC 15K	SGL 90L/ HPMC 15K	HPMC 15K
		F4-CIP	F10-CIP	F16-CIP
*C_max_* (µg/mL)	0.86 ± 0.24	2.49 ± 0.20 *^,#^	2.77 ± 0.38 *^,#^	1.60 ± 0.36
*T_max_* (h)	1.33 ± 0.29 ^#^	1.33 ± 0.29 ^#^	1.33 ± 0.29 ^#^	2.0 ± 0.0
MRT (h)	3.71 ± 0.71	2.85 ± 0.53	3.78 ± 0.91	4.18 ± 0.99
AUC_0−t_ (µg/mL/h)	3.30 ± 0.63	7.87 ± 1.52	12.78 ± 3.25 *	8.16 ± 1.50
AUC_0−∞_ (µg/mL/h)	4.72 ± 0.85 ^#^	8.71 ± 1.32	14.33 ± 4.28 *	13.21 ± 4.33 *
Relative bioavailability ^1^ (%)	−	238.35 ± 45.94	387.33 ± 98.43	247.34 ± 45.46

^1^ Relative bioavailability = AUC_0–t (*sf*GRDDS formulations)_/AUC_0–t (commercial tablets)_ × 100%. * indicates significant difference (*p* < 0.05) compared to the commercial tablets group. ^#^ indicates significant difference (*p* < 0.05) compared to the F16-CIP group. Data represent mean ± standard error of the mean (*n* = 3 for each group).

## Data Availability

The datasets used and/or analyzed in the current study are available from the corresponding author upon reasonable request.

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
