# Peer review of "Development of a Swellable and Floating Gastroretentive Drug Delivery System (sfGRDDS) of Ciprofloxacin Hydrochloride"

_pharmaceutics, 2023, doi:10.3390/pharmaceutics15051428_

Round 1

Reviewer 1 Report

The manuscript „Development of Swellable and Floating Gastroretentive Drug Delivery System (sfGRDDS) of Ciprofloxacin Hydrochloride with Hydrophobically Modified Hydroxypropyl Methylcellulose: Formulation, Physicochemical Characterization, and In vivo Pharmacokinetics” handed in by Liang et al. has a clear structure but there is the need for some improvement. Please find my comments below:

-          General: the entire document has to be formatted to avoid page breaks in tables etc.

-          Line 26: in the abstract it is not clear what is meant by “15K”. Please rephrase or include more information.

-          Abbreviation HPMC: once the abbreviation is introduced in the document, please use it throughout. Introduction was in line 61, so please change to HPMC in lines 71, 72, 523.

-          Line 98: please rephrase the sentence to: “And CIP is mainly absorbed in the proximal gastrointestinal tract”.

-          Table 1 and 2: please add the information that this is the composition and total weight for 1 tablet.

-          Lines 155 ff: it is not clear how the times were measured here, please add this information in the manuscript.  

-          Lines 198 and 254: the R2 should be a capital R, as it will be used later in the text as well.

-          Lines 294 ff: there, the study discussed considers the diameter of the tablets after 12 h. In the current study experiments were made until 8 h. Can these results be compared with each other? Is there some literature regarding 8 h to enable a comparison? Please comment on this. Line 294: please use “12 h”

-          Lines 305 and 306: please use only the abbreviations of FLT, they were already introduced above.

-          Table 4: what does 0 and x mean? Please add an explanation.

-          Line 361: please rephrase the end of the sentence “resulting in to decrease in the initial…”

-          Figure 4h: the text does not refer to this curve. Please include this in the manuscript.

-          Table 5: is this table really necessary and does it add value to the manuscript. Could the information regarding MDT and DE be added in figure 4?

-          Line 419: Please explain why it was chosen that a fit below R2 = 0 is a poor fit. Do not only refer to literature and explain in more detail because this is not common use.

-          Please rephrase the following sections:

o   Line 479 “showed significantly higher”

o   Line 481: “no statistically different”

o   Line 484: “had significantly higher than”

o   Line 486: “exhibited significantly higher than”

Reviewer 2 Report

1. Elaborate in detail about the method of preparation of Tablet with reference.

2. Modified HPMC is synthesized or marketed company product. 

3. Few formulation must be prepared with HPMC only to check the difference in properties.

4. Why the release media 0.1 Hcl selected, it must also be with gastric pH.

5. How much was loading dose.

6. How the optimum formulation selected.

7. What are the significant findings of the study.

Reviewer 3 Report

This is a nice and well written report describing the GRDDS tablets of ciprofloxacin HCL comprised of SGL and HPMC. The authors mentioned that the SGL-HPMC based GRDDS tablets have the potential to enhance the bioavailability of antibiotic CIP.

The study is well designed, and the results are clearly displayed. However, there are some gaps that should be addressed.

1)      The chemical structures (Figure 1) of SGl and CIP should be with identical font size. Also keep the bond width identical for both the structures.

2) It will look visually appealing if authors include the time based floating and swelling behavior of SGL-HPMC tablets (atleast for batch F3-CIP, F4-CIP, F9-CIP, F10-CIP, F15-CIP, F16-CIP and F21-CIP.

Reviewer 4 Report

The manuscript on the swellable and floating formulation of ciprofloxacin contains noticeable results. The current version is almost suitable for publication, and the referee could find only some minor issues.

- The title is unnecessarily long, especially regarding the Keywords section. The use of hydrophobically modified HPMC is not necessary for the title. A shorter version, like the 'Development of a Swellable and Floating Gastroretentive Drug Delivery System (sfGRDDS) of Ciprofloxacin Hydrochloride' can correctly characterize the content.

- In the Keywords section, the repetition of drug delivery system and hydrophobically modified hydroxypropyl methylcellulose overemphasize an already mentioned information. Of course, after shortening the title, the latter can remain in the Keywords.

- In lines 41-43, the authors associating the short plasma half-life of CIP with oral administration is not necessarily correct. The short plasma half-life is obviously concentration-dependent, as all chemical/biochemical reactions. The poor oral bioavailability makes a loose connection between oral administration and half-life only, but the sentence suggests a stronger statement. Please reformulate that sentence.

Round 2

Reviewer 2 Report

Accept

Author Response

Dear Reviewer,

I am writing to express my sincere appreciation for your comments and suggestions. Your input has been invaluable in improving our work.